# Measuring Organizational Culture in Hotels, Restaurants and Travel Agencies in Montenegro

Olivera Simovic [1], Miha Lesjak [2], Đurđica Perović [1] and Eva Podovšovnik [2,*]

1    Faculty of Tourism and Hotel Mangement, University of Montenegro, Old Town 320, 85330 Kotor, Montenegro
2    Faculty of Tourism Studies, University of Primorska, 6000 Koper, Slovenia
*    Correspondence: eva.podovsovnik@fts.upr.si

**Abstract:** Organizational culture has a strong influence on the management of the organization because cultural patterns are at the core of all human behavior and actions. The aim of this paper is to determine the character of the dominant models of organizational culture in the tourism and hospitality sector of Montenegro. In this study, a field survey questionnaire was used, among employees of hotels, restaurants and travel agencies in Montenegro. In this study, we examined specifically how the characteristics of the company affect the perception of its organizational culture. Furthermore, in this study we also examined how the characteristics of respondents affect the perception of organizational culture. Research hypotheses were tested with CHAID analysis, using IBM SPSS Statistics 26. Results show that the largest number of hotel and catering companies in Montenegro have an organizational culture of the clan and the second most represented culture is the culture of hierarchy. The main finding from the conducted research is the fact that the perception of the dominant type of organizational structure is significantly influenced by the type of tourist company and geographic distribution. The obtained results show that in hotels in the southern part of Montenegro, hierarchy culture is predominant, while in restaurants and travel agencies in central and northern regions it is the clan culture that dominates. The practical contribution implication of the study is in creating knowledge that can be used by managers of tourism companies in Montenegro to create a corporate strategy, recognizing that organizational culture has become an important aspect for senior management, i.e., managing the company and its development.

**Keywords:** organizational culture; sustainability; tourism; hotels; restaurants; travel agencies

## 1. Introduction

Organizational culture draws researchers' attention as it is a fundamental factor which influences organizations' performance [1,2].

Management researchers support the attitude that the "glue" that holds all the components of the organization together, giving it an internal strength and the ability to adapt to turbulent changes in the environment, is a system of assumptions, beliefs, values, norms and attitudes built by employees through the organizational practice itself; in other words, what makes up its organizational culture [3].

If we want to understand the causes, the forms and the consequences of the behavior of people in the organization, we must know its culture. In practical terms, organizational culture describes the environment in which people work and the values on which their business is based [4].

So far, several attempts have been made to perceive the state of the current empirical literature in the research of the relationship of hotel companies' performance and types of organizational culture. Tavitiyaman [5] states that organizational culture is characterized by different factors that have positive or negative effects on an organization's performance, whether those are financial, market or organizational performances. Mui and Cheok

provided one of the first systematic reviews of the studies that examine the relationship between the size of hotel companies and predominant type of organizational culture on the micro level [6]. Empirical research that analyses the type of organizational culture and high category hotels, with special focus on green hotels as innovative segment on the market, has been carried out by more comprehensive examination [7]. For example, a study in Mexico, Oxaco, on the sample of 130 hotels, where the author uses a competing values framework (CVF) matrix by analyzing the relationship between multiple types of organizational culture, including hierarchy culture, clan, adhocracy and market, and different types of eco-innovation. Research results suggest the importance of the company's size and adhocracy culture as crucial eco-innovation determinants. Dawson and Abbot [8] presented a general and comprehensive review of all determinants in hotel companies that show that organizational culture typologies have been designed to improve a company's performance, as organizational culture affects the relationship between the hotel owner and employees by building their trust and influencing employees' dedication and performance. Furthermore, Hakami's [9] study suggested that organizational culture and total quality system play an essential role in generating positive effects on a company's performance. Similarly, Oriade et al. [10] have found a correlation between a tourism company's development level in a certain region, on one side, and perception of organizational culture, on the other side.

The previous literature mainly looks at the organizational culture as a complex system on the company's level, while a small number of studies look at the phenomenon of organizational culture on an individual level [11]. Ramadistaa and Kismono [12] found that organizational culture significantly influences a company's business performance through the practice of human resources management. Perception and experience are basically considered phenomena of subjective character and are defined by an individual's characteristics [13]. A considerable part of the literature dealt with examining of the antecedents of organizational culture [14]; however, there are only a few studies that examine the influence of individual characteristics on organizational culture. In that regard, Sommer et al. [15] emphasized the importance of an individual's characteristics, such as their workplace, mandate and age, pointing out that those characteristics significantly influence and determine one's dedication to an organization. Furthermore, authors discovered that older employees who have been working on higher positions for a long time show more dedication to the organization [16]. In addition, previous studies show a strong connection between the perception of organizational culture, as an important subsystem of the overall culture, and dedication to an organization [17,18].

Previous studies mainly highlighted the examination of correlation between organizational culture and a company's performances in the context of financial outcomes and economic efficiency [19]. However, this study is focused on examination of connection between organizational culture, on one side, and a company's performances and individual characteristics as a key predictor of dominant type of organizational culture, on the other side.

In order to examine this connection, we used the competing values framevork (CVF) of organizational culture, the most frequently used framework in empirical research, which has been developed by Cameron and Quinn [20,21]. The proposed framework differentiates between four types of organizational culture: clan culture, hierarchy, market and adhocracy [21].

Clan culture favors flexibility and autonomy over stability and control [22]. This culture is characterized by an organization that is a very friendly place for its members. This implies strong relationships between members, where friendship, mutual support and teamwork are highly valued. At the same time the leader is very respected, and their role is twofold: as a leader and as a supervisor. In other words, clan culture is by definition characterized by tradition, loyalty, teamwork, personal commitment and close interpersonal relationships.

Its opposite, hierarchy culture, is characterized by strictly formalized rules and procedures aimed at achieving greater efficiency [23]. Keeping to the best practices, controlled

processes and constant monitoring is considered necessary for productivity and success in this type of organizational culture. Hierarchy is a type of organizational structure in which items are ranked according to their degree of importance. In this type of organization, hierarchy depends on structure, rules and control which are carried out from the top to the bottom in order to conduct business practice and activities. In the hierarchy culture, the emphasis is on the inner orientation. An organization with this type of organizational culture is a formalized and structured place, while people are united by keeping to the same rules and procedures.

Adhocracy culture refers to a flexible, adaptable and informal form of organization, which is characterized by entrepreneurial spirit and innovation in problem solving [24]. This type of organizational culture makes the organization a dynamic, creative and entrepreneurial place. People in the organization are connected by a desire to experiment and try new things.

Within the market organizational culture, the organization is oriented towards achieving the best possible result [21]. Competitiveness is expected from people, as well as goal- and result-oriented behavior. Leaders encourage hard work, achieving results and competition among employees. People are held together by the desire to win. It is a culture that has a distinctly competitive character. Everyone competes here: employees in the organization among themselves and the organization with other competing organizations in the market. Rivalry among members of the organization is encouraged.

Theory predicts that a company's organizational culture depends on its type and size [1]; however, none of these relevant sources include a country's regional specificity as a variable. This study expands current conclusions by placing focus on its geographic position, i.e., geographic distribution of tourism companies, as a powerful variable of predicting the dominant type of organizational culture. Tourism economy is characterized by different levels of development based exactly on a destination's regional specificity [25] that has immediate effect on the dominant type of organizational culture in tourism companies depending on regional affiliation.

In this study, we offer a synthesis of the current empirical research which examines the perception of the organizational culture in tourism economy, expanding the current literature reviews in several ways. Firstly, we summarized theories that are used as the basis for empiric research. Secondly, we included tourist companies as well as hotels, restaurants and tourist agencies, and by doing so we made a comprehensive approach to the research of organizational culture in the tourist economy as a whole. Additionally, we covered empirical studies on the micro level, providing more detail on how a company's characteristics and type influence the dominant type of organizational culture, not leaving out individual characteristics of respondents and regional distribution of tourist companies. In addition, we also included the macro level, in regard to tourist destination, giving a more general image of the dominant type of organizational culture in the Montenegrin tourism industry, which is characterized by strong economic transition [26]. Montenegro offers an ideal working environment for this study due to two important reasons. Firstly, two decades of economic reform were reflected in the change of ownership and organizational structure of tourism companies, which have been transformed from self-governing workers' organizations into different types of private enterprises with domestic and foreign capital investments [27]. This made Montenegro a very attractive market for many multinational companies. Secondly, due to inadequate knowledge of organizational culture in developing economies, this study contributes to findings from previous examinations. Finally, the analysis resulted in important proposals for future examinations of this topic and identified some principal methodological issues.

The paper is organized as follows: introduction, theoretical framework, literature review, which includes the relationship between organizational culture and demographic characteristics and relationship between organizational culture and the characteristics of the company in the tourism and hospitality sector, then we develop research hypotheses and the research instrument. In the following, a description of the research design is presented,

then we present describe the sample and research results. Discussion and conclusion with limitations and recommendations for future development are presented at the end of the paper.

## 2. Literature Review

### 2.1. Relationship between Organizational Culture and Demographic Characteristics

The concept of organizational culture originated in the early 1980s and made a major contribution to the theory and practice of management [28] and business management [29]. Most authors in the field of management view organizational culture as a part of the package of management tools available to managers to raise the level of effectiveness of organizations which they manage [30]. According to Azeem [31], organizational culture is considered to be the most important organizational capital. Organizational culture is a set of norms and values which are applied in the organization [32] and which determine the identity of the organization [33].

According to Schein [34], organizational culture is: "a pattern of shared basic assumptions that the group has invented, discovered and developed in learning to cope with its problems of external adaptation and internal integration, that has worked well enough to be considered valid and, therefore, to be taught to new members as the correct way to perceive, think, and feel in relation to those problems".

Schwartz and Davis's definition includes the element of expectation into the organizational culture and stresses the importance of individual's characteristics: "Culture is a model of beliefs and expectations shared by the members of an organization" [35]. Those beliefs and expectations produce norms which strongly shape the behavior of each individual and groups in an organization. Employees' expectations strongly influence the subjective perception and significantly determine behavior, especially in regard to satisfaction, devotion and personal progress. The prospects of successful development or indication of an upcoming crisis and possible end to an organization's businesses can cause completely different behavior in an individual or a group of employees, which largely depends on their personal characteristics, and which range from an increased work energy and enthusiasm to unwillingness and conflict behavior. Expectations shift the focus from organizational culture as it is, to organizational culture as it should be according to the perception of organization's members.

Cultures can vary significantly within and between organizations. They can get the best out of people and create the right and functional work environment, or they can pull out the worst in people and create a dysfunctional environment filled with stress and tension [36]. In general, the theory of organizational culture assumes that organizational culture exerts its influence through shaping the behavior of the members of the organization [2,37]. Therefore, organizational culture is a necessary organizational condition to improve the work engagement and performance of the members of the organization.

In the context of hospitality, the development of organizational culture has become very important due to the intangible nature of services in hospitality, and the subjective perception of the quality of the service provided [38]. It should be kept in mind that the organizational culture in tourism is specific in the way that the product and service are integrated and create a unique experience for the guest [8]. Accordingly, employee characteristics are considered crucial in creating that experience [39]. According to Justwan, Bakker and Berejikian [40], organizational culture influences the relationship between hotel owners and employees by building trust between them and the impact on organizational commitment and employee performance. This additionally highlights the importance of individual characteristics and the perception of organizational culture.

### 2.2. Relationship between Organizational Culture and Characteristics of the Company in the Tourism and Hospitality Sector

In a theoretical and practical sense, organizational culture takes a special place in hospitality service as it considerably influences the improvement of tourism company

performances [20]. Due to clients' increased needs and demands, as well as the pronounced competitiveness of the global business environment, companies have started to look for new ways to respond to more demanding conditions in the market and to create additional value for their products and services [41]. Organizational culture is the key determinant of hotel policy in the introduction and implementation of management practices, business processes and strategies, in regard to the fact that they are founded on the beliefs and values of the organization. In line with the literature [42,43], organizational efficiency depends on several factors: the size of an organization, its structure and its age. Furthermore, Hofstede [44] supplements the present knowledge by introducing a geographic variable, i.e., geographic expansion and regional distribution on the national level, as an important determinant of the dominant type of organizational culture. In this regard, Cerović and Tomašević [45] examined symbolic and cognitive elements of organizational culture in 121 hotels across Europe. They came to a conclusion that the quality of services offered to hotel guests will be determined depending on the type of organizational culture that dominantly prevails; accordingly, organizational culture directly influences an organization's success.

The organizational culture of tourism and hospitality facilities, from the perspective of users of tourist services, is primarily manifested through the quality of the service that these facilities provide to visitors [33]. This is an important reference for the user when deciding which hotel to stay in or which restaurant to choose when visiting a tourist destination. The quality criterion is equally important for travel agencies as the main promoters of tourist facilities which they include in their travel arrangements.

The best tourism and hospitality organizations are recognized precisely because their organizational culture is practically a synonym for a quality culture, which is guided by the belief that the most important thing for the business success of the organization is that everyone in the organization is committed to achieving a high level of hospitality service [46]. Researchers have developed many theories and models of organizational culture, mainly by exploration of manufacturing sector [47]. Those theories and models have been used in many studies on the relationship between organizational culture and performances [48].

Neither of these relevant literature reviews focus on the underdeveloped and transition countries, nor do they include a country's regional specificity as a variable. In this work, we are trying to deal with these shortcomings because tourism economy is characterized by different levels of development, based exactly on a destination's regional specificity [28] that has immediate effect on the dominant type of organizational structure in tourism companies depending on regional affiliation. Additionally, we covered empirical studies on the micro level, providing more detail on how a company's characteristics and type influence the dominant type of organizational culture. We joined tourist companies as well as hotels, restaurants and tourist agencies, and by doing so we made a comprehensive approach to the research of organizational structure in the tourist industry.

### 2.3. Sustainable Organizational Culture

Modern business tendencies introduce the concept of corporate sustainability, which is gaining increasing attention in the theory and practice of business management. In that regard, Deloitte defines corporate sustainability as a comprehensive approach that is focused on the creation and maximizing of long-term economic, social and ecological values [10]. Comprehension of corporate sustainability derives from the general term of sustainability, which was generated due to social, political and academic influences [49]. Most studies [50] show that corporate sustainability refers to an organization's social responsibility [51] or ecological concern [52]. However, there is still insufficient knowledge on how to apply corporate sustainability to organizational practice [53].

According to Haris and Krejn [54], in order to establish corporate sustainability, it is necessary to establish an organizational culture oriented toward sustainability. In this respect, it is strongly believed that without developing a stable and green organizational culture it would not be possible to improve the sustainable performance of an enterprise [55].

Corporate sustainability is a multifaceted concept that requires organizational changes and adaptations on different levels, in compilation with levels of organizational culture. On the surface level, corporate sustainability is reflected through technical solutions or the training of employees [56]. On the values level, corporate sustainability is achieved by the adoption of ethically responsible values, through changes in the values and beliefs of the employees [57]. On the main level, basic presumptions related to understanding and accepting ecological systems are changing [58]. Sharma et al. proved in their study that compatibility of organizational culture with the environment can improve company's performances [59]. Their results are compatible with the results of Cabral and Dhar, and Zameer et al. [60,61], who claim that organizational culture focused on sustainability has many advantages in improving a company's performance, including help with productivity improvement [62], costs minimizing [63], protection of the environment and ecological awareness of the employees in an organization [64], and improvement of long-term financial performances [65]. In that regard, this places the organizational structure as the most important determinant in forming of sustainability awareness that further leads to improvement of business performances. Schein's study shows that the culture within an organization is shaped by the values and beliefs that employees hold on to in practice [2].

However, some authors such as Visser, Oriade, Osinaike, Aduhene, Wang and Brauer [10,65,66] noticed that most of theoretical and empirical research in this field comes from developed western countries. Erdogan and Baris stress that managers lack the necessary knowledge and interest to fulfill the main goals of social and ecological responsibility, especially with regard to developing countries, and because of that they propose effective learning and suitable gaining of knowledge to be able to understand and work on sustainability issues [67].

Analyzing recent studies, it has been noticed that managers in the hotel industry have started to follow trends related to environmental issues [68]. Cop and his associates discovered that green transformation leadership has a positive effect on green work engagement in hotel enterprises [69]. In this sense, green organizational culture can help in the effective conduct of green practices and behaviors.

In his study, Gürlek explained that culture imposes a pressure on individuals, making them act and behave according to cultural values and behavior norms [70]. Based on the previous literature, we realize that organizational culture is one of the most important determinants in forming of ecological awareness, along with different sociopsychological variables. If an organizational culture promotes the value that employees should act in an ecologically conscious way, that will encourage their behavior to be in accordance with the values of an organizational culture that is led by the attitude that environmental protection and sustainable management through ecological responsibility are the essential values of the company [71]. Hence, organizational cultures focused on sustainability can encourage employees and make them show responsible social and ecological behavior [72].

### 2.4. Hypotheses Development

The previous literature mainly looks at an organizational culture as a complex system on the company's level, while a small number of studies look at phenomenon of organizational culture on individual level, starting with an individual [11]. A considerable part of the literature dealt with examining antecedents of organizational culture [14]; however, in our opinion, there are only a few studies that examine the influence of individual characteristics on organizational culture. Guided by previous studies that examined the influence of an individual's characteristics, including their age, education level and work experience, on their dedication to organization [18], in this study we tried to examine the correlation between individual characteristics and perception of organizational culture.

Organizational culture, as a socially constructed context, depends on multiple demographic factors, such as gender, age, working experience in general, level of education and position within an organization [73]. OReilly and Chatman [11] suggest that those values and beliefs that an organization's employees gravitate to condition their behavior and inter-

action with the organization's members, as well as the decision-making process in solving the challenges an organization faces. In general, differences in demographic characteristics are in positive correlation to individual development and perspectives of employees of a different background and age that further leads to identification with values in organization and better organizational performance [74]. Some authors point out positive effects of demographic heterogeneity through the prism of values, behavior norms and individual characteristics [15], while other authors consider that heterogeneous working groups face limitation in communication and social interaction, which causes less commitment and connection within an organization [75]. Several studies of the hospitality sector discovered that demographic factors such as gender, age, level of education and work experience influence employees' preferences and their perception of organizational culture [8,76]. For example, Sayli et al. [73] discovered that the perception of the organizational culture of employees in hospitality tourism companies largely depends on age, education, work experience and interpersonal relationships. Belias and Koustelios [75] revealed employees' gender to be a significant antecedent of not only dominant but also preferred organizational culture. Furthermore, results show that women prefer a family-like working environment which tends to a clan culture, while men prefer competitive hierarchy context, as seen in market culture and hierarchy culture. Apart from gender, level of education corresponds to a company's organizational culture. More precisely, the role of education is crucial for conceptualization of organizational culture through a system of values and practice [77]. In this regard, the study of Tsui et al. [78] shows that employees with higher levels of education perceive their organization as hierarchical and show an affinity for a formalized and structured working environment while, on the other side, employees with lower levels of education prefer a working environment characterized by clan organizational culture, i.e., taking care of people and a family atmosphere. Differences in demographic characteristics, including gender, age, work experience or level of education, can significantly influence employees' behavior, primarily related to interpersonal relationships, socialization and devotion to the organization.

Guided by previous studies that examined influence of individual's characteristics, including their age, education level and work experience, on their dedication to organization [18], in this study we tried to examine correlation between individual characteristics and perception of organizational culture in tourism companies in Montenegro. Previous studies show a strong connection between the perception of organizational culture, as an important subsystem of the overall culture, and dedication to an organization [16]. As such, the first research hypothesis can be developed:

**Hypothesis 1 (H1).** *Respondents' characteristics (gender, age, highest level of education, years of working experience and type of employment) influence the perception of organizational culture.*

Only a few organizational culture studies consider variables such as the size of a company and geographical location as factors that shape organizational structure in the hospitality sector [79]. Related to this, Konovalova et al. [25] stressed the important correlation between organizational culture and tourism company characteristics, pointing out two key contextual variables, such as the size of a company and geographical location, as significant determinants of organizational culture. Prajogoand McDermott [80] came to the same conclusion, stating that the comparison of small and big companies will expand our perception of organizational culture and its dominant types. Previous studies showed that large companies are more likely to have a formal and stable structure [1]. Denison and Spreitzer [81] stated that organizations have tendency to become more formal as their size grows, which lends itself to a hierarchy organizational culture. On the contrary, small companies tend to act informally and flexibly, possessing a clan culture [1]. Geographic factors significantly condition the level of tourism development on the national level, which is reflected in the type of organizational culture in tourism and hospitality companies [44]; in a sense, regions characterized by highly developed tourism show organizational culture with a high level of formality and bureaucracy, while, on the other hand, regions with

underdeveloped tourism show informal structure and care more about people as opposed to competitiveness and profit [28].

Bearing in mind all of the above, it is obvious that a company's characteristics represent an important determinant of organizational culture and, with that, the overall progress of an organization. The type, size and regional affiliation of a company that have key roles in the creation of specific values and the perception of organizational culture [82]; therefore, measurement of organizational culture is a logical outcome and a necessity in terms of creating an adequate strategy for tourism sector companies and their business and performance.

According to the theoretical background mentioned above, the second research hypothesis can be developed:

**Hypothesis 2 (H2).** *Companies' characteristics (type of company and region of the company) influence the perception of organizational culture.*

## 3. Research Design

In this section, the research design utilized will be presented.

### 3.1. Research Problem

With regard to the objectives of the research, and based on the studied literature, we have defined the research question, i.e., the general question of work from which results the basic hypotheses that are the subject of testing in the empirical part of the research. Having in mind that the Montenegrin economy is predominantly focused on the tourism sector, within which the tourist offer of domestic hotel and hospitality companies has a particularly important role [83], we focused the research question on the analyses of current trends in organizational culture in the surveyed companies.

### 3.2. Research Instruments (Variables and Measurement)

As a typology of organizational culture, we chose the classification of Quinn and Cameron [84]. It is based on a theoretical model known as the competing values framework (CVF) and Organizational culture assessment instrument (OCAI), which distinguish four types of organizational culture: clan, hierarchy, market and adhocracy.

The following research model will be tested (see Figure 1).

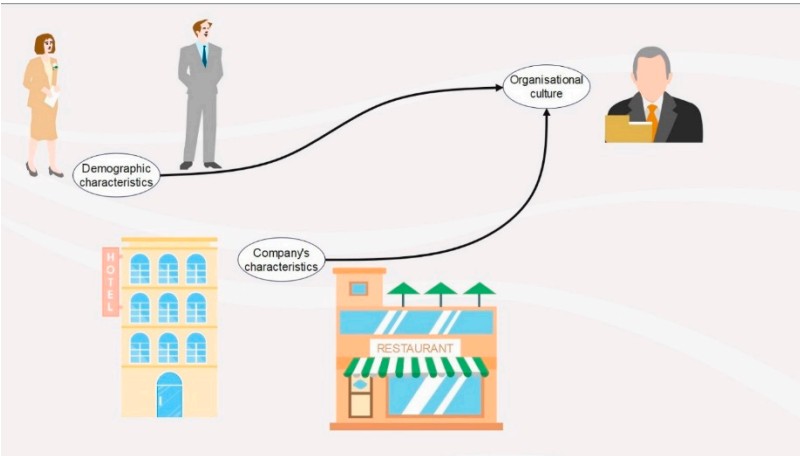

**Figure 1.** Research model.

In order to test the mentioned research model, CHAID analysis was used, as categorical variables (in our case gender, geographical location of the company, type of company, type of employee) can be inserted as independent variables in the model, as well as the dependent variable (the dominant type of organization). This analysis offers a graphical

presentation of the obtained groups and their most distinctive characteristics. All statistical analyses were performed using IBM SPSS Statistics version 26.

Considering the fact that the research was carried out in order to analyze the dominant type of organizational culture at the level of organizations, a survey was conducted. The full questionnaire can be found in Supplementary Materials. The survey questionnaire was composed of two parts:

1.  Cameron and Quinn's questionnaire (OCAI) for diagnosing the organizational culture [21], with questions related to the dominant characteristics of the organization, leadership in the organization, people management, unity of the organization, strategic focus and criteria of success. This questionnaire belongs to the typological group of questionnaires, and thus organizational culture is classified into one of four types: clan, hierarchy, market or adhocracy. OCAI is one of the most influential and widely used models in the field of organizational culture research compared to other models and scales [46].

2.  Demographic characteristics: gender, age, level of education, years of work experience, type of employment, as well as company characteristics, i.e., type of company and regional affiliation.

### 3.3. Target Population, Sample Frame, Sampling

The target population for the mentioned research were tourist companies that:

1.  had gone through a process of change over the past decade;
2.  were of different sizes;
3.  were of different ownership structure;
4.  had different performances;
5.  were distributed in all three tourist regions of Montenegro.

When it comes to companies that own several tourist facilities, we conducted the survey in the most representative facility of that company (e.g., hotel "Splendid" of the company Montenegro Stars Hotels Group).

The Central Register of Economic Entities of Montenegro had in its database at the beginning of the survey a total of 1228 business entities registered under the code 5610 (tourism activities), so these 1228 companies represent the population from which we obtained the survey sample. Companies are classified according to the activity they perform in three basic categories: hotel companies (423), restaurant companies (470) and companies that act as travel agencies (335).

Due to the nature of the research we conducted, the relevant research companies were only those that are older than five years, and their number was: 300 hotels, 260 restaurants, and 142 travel agencies. Additionally, as tourism is one of the priority and leading branches of the economy in Montenegro, and there is an obvious difference in representation by regions, an additional categorization by region was considered relevant for the described research, which gave the following situation: southern region of Montenegro: 287 hotels, 225 restaurants, 179 travel agencies; central region of Montenegro: 82 hotels, 167 restaurants, 142 travel agencies; and northern region of Montenegro: 54 hotels, 78 restaurants, 14 travel agencies.

Regarding the planned sample size, we adopted the view of the authors who consider representative a sample whose size is 10% of the population if its size allows reliable data analyses and testing the significance of the differences in estimation [85,86]. We also took into account that the size of the sample was chosen depending on the subject, purpose, usefulness and credibility of the research conducted with the available resources and in the available time [87]. For the reason of greater precision, we applied the standard statistical formula for calculating the research sample, which is:

$$S = \frac{\frac{z^2 \cdot p(1-p)}{e^2}}{1 + \frac{z^2 \cdot p(1-p)}{e^2 \cdot N}}$$

where *S*—sample size, *z*—mark for given confidence interval, *p*—percentage, *e*—margin of error and *N*—population size. *Z* = 1.96 (for the confidence interval of 95%), *p* = 0.5 (5%), *e* = 0.08 (8%) (we increased the margin of error from the standard 5% to 8% so that the size of the representative sample would be within 10% of the total number of tourist companies in Montenegro.) and *N* = 1228 (total number of tourism companies in Montenegro).

Based on this formula, a representative sample size of 134 companies was formed.

Due to the nature of the research being conducted, additional stratification of the sample was performed, by type of tourism companies and tourist regions in Montenegro, which resulted in the sample structure outlined in Table 1 (stratified random sampling is applicable if the population from which the sample is selected does not represent a homogeneous group [88]. By applying the method of proportional allocation, the size of the stratum of the sample population is proportional to the size of the stratum of the population [89] which ensures the generalization of conclusions after the survey [90]).

**Table 1.** Representative sample structure.

| Type of the Object | Montenegro | Southern Region | Central Region | Northern Region |
|---|---|---|---|---|
| Hotel | 47 | 27 | 13 | 7 |
| Restaurant | 52 | 25 | 15 | 12 |
| Agency | 35 | 18 | 15 | 2 |
| **Total** | **134** | **70** | **43** | **21** |

*3.4. Data Collection Technique*

The field survey was conducted in the period from May 2018 to September 2019. The questionnaires were distributed in printed form. The survey was anonymous, with the obligatory stamp verification of the companies covered by the survey. These companies were previously sent a request to participate in the research, with a survey questionnaire and an explanation that no confidential company data or financial indicators were required. Then, with the approval of the company's management, a survey was scheduled, after which, with their help and support, the survey was started. Questionnaires were distributed by region, and for simpler organization and systematicity, the survey was conducted sequentially, region by region.

The questionnaires were handed out depending on the number of employees and the type of tourism companies (hotels, restaurants or travel agencies), which were determined before scheduling the survey. The range was from five to 33 respondents in hotels, four to 30 respondents in restaurants and two to 21 respondents in travel agencies. Respondents who filled out the questionnaire were selected by random selection from the employees who were on shift at the time of the survey questionnaires. Depending on the agreement with the management of the organization, some tourism companies returned the questionnaires on the same day they were filled out, while others did so within the seven days given for their completion. A total of 2050 survey questionnaires were distributed, 1992, or 97%, of which were returned. After discarding incomplete or incorrectly completed questionnaires, 1523 of them were taken into account for analysis.

*3.5. Description of the Sample*

First, the description of respondents is presented (see Table 2).

As it can be seen from Table 2, 51.2% of the respondents were male and 48.8% were female. 33.2% of them were aged between 36 and 45 years, 30% of them were aged between 26 and 35, 18.4% of them were aged below 25 years, 16% of them were aged between 46 and 55 years and 2.4% of them were older than 55 years. Half of the respondents (51.1%) had elementary or high school completed, 35.4% had a college or bachelor degree, and 12.9% of them had a postgraduate degree (master or doctoral). About one third (34.6%) of respondents had between 11 and 20 years of working experience, 31.9% of them had

between six and 10 years of working experience, 17% of them had less than five years of working experience, 13.2% of them had 21 to 30 years of working experience and 3.3% of them had more than 30 years of working experience. 68.1% of respondents were operative workers, while 31.9% were managerial workers.

**Table 2.** Description of the sample.

| | f | f% |
|---|---|---|
| **Gender** | | |
| Male | 781 | 51.2 |
| Female | 744 | 48.8 |
| **Age** | | |
| Below 25 | 281 | 18.4 |
| 26 to 35 | 457 | 30.0 |
| 36 to 45 | 506 | 33.2 |
| 46 to 55 | 244 | 16.0 |
| 56 and more | 37 | 2.4 |
| **Highest level of education** | | |
| Elementary or high school diploma | 789 | 51.7 |
| College or bachelor degree | 540 | 35.4 |
| Postgraduare degree | 196 | 12.9 |
| **Years of working experience** | | |
| Up to 5 years | 259 | 17.0 |
| 6 to 10 years | 486 | 31.9 |
| 11 to 20 years | 528 | 34.6 |
| 21 to 30 years | 201 | 13.2 |
| More than 30 years | 51 | 3.3 |
| **Type of employee** | | |
| Managerial | 486 | 31.9 |
| Operative | 1037 | 68.1 |

In Table 3, the characteristics of the company are presented.

**Table 3.** Characteristics of the company.

| | f | f% |
|---|---|---|
| **Type of company** | | |
| Hotel | 816 | 53.5 |
| Restaurant | 528 | 34.6 |
| Travel agency | 181 | 11.9 |
| **Region of the company** | | |
| Central | 491 | 32.2 |
| South | 862 | 56.5 |
| North | 172 | 11.3 |

About half of the respondents worked in a hotel (53.5%), 34.6% in a restaurant and 11.9% in a travel agency. 56.5% of respondents worked in the south region, 32.2% in the central region and 11.3% in the north region.

## 4. Results

In this section, the results are presented.

### 4.1. Dominant Organizational Culture

The OCAI questionnaire was developed by Cameron and Quinn, the authors of the CVF model which distinguishes four types of organizational culture: clan culture, hierarchy culture, adhocracy culture and market culture [88]. Their purpose is to help leaders in organizational changes identify their current and desired culture. The questionnaire contains six questions which represent six essential components of organizational culture (see Table 4):

1.  dominant characteristics—what are the main characteristics of the organization;
2.  organizational leadership—what are considered to be the abilities of the leader of the organization;
3.  management of employees—which management methods are applied to employees in the organization by management;
4.  organization glue—what is it that keeps the organization together, i.e., how the organization consolidates itself;
5.  strategic emphases—what the organization is predominantly focused on in its work;
6.  criteria for success—how is success defined in an organization.

**Table 4.** The OCAI questionnaire.

| Dimension of Organisational Culture | | Items |
|---|---|---|
| Dominant characteristics | A1 | The organization is a very personal place. It is like an extended family. People seem to share a lot of themselves. |
| | B1 | The organization is a very dynamic and entrepreneurial place. People are willing to stick their necks out and take risks. |
| | C1 | The organization is very results-oriented. A major concern is with getting the job done. People are vey competitive and achievement-oriented. |
| | D1 | The organization is a very controlled and structured place. Formal procedures generally govern what people do. |
| Organisational leadership | A2 | The leadership in the organization is generally considered to exemplify mentoring, facilitating, or nurturing. |
| | B2 | The leadership in the organization is generally considered to exemplify entrepreneurship, innovation, or risk taking. |
| | C2 | The leadership in the organization is generally considered to exemplify a no-nonsense, aggressive, results-oriented focus. |
| | D2 | The leadership in the organization is generally considered to exemplify coordinating, organizing, or smooth-running efficiency. |
| Management of employees | A3 | The management style in the organization is characterized by teamwork, consensus, and participation |
| | B3 | The management style in the organization is characterized by individual risk taking, innovation, freedom, and uniqueness |
| | C3 | The management style in the organization is characterized by hard-driving competitiveness, high demands, and achievement. |
| | D3 | The management style in the organization is characterized by security of employment, conformity, predictability, and stability in relationships. |

**Table 4.** *Cont.*

| Dimension of Organisational Culture | | Items |
|---|---|---|
| Organisation glue | A4 | The glue that holds the organiztion together is loyalty and mutual trust. Commitment to this organization runs high. |
| | B4 | The glue that holds the organiztion together is commitment to innovation and development. There is an emphasis on being on the cutting edge. |
| | C4 | The glue that holds the organiztion together is the emphasis on achievement and goal accomplishment. |
| | D4 | The glue that holds the organiztion together is formal rules and policies. Maintaing a smooth-running organization is important. |
| Strategic emphasis | A5 | The organization emphasizes human development. High trust, openness, and participation persist. |
| | B5 | The organization emphasizes aquiring new resources and creating new challenges. Trying new things and prospecting for opportunities are valued. |
| | C5 | The organization emphasizes competitive actions and achievement. Hitting stretch targets and winning in the marketplace are dominant. |
| | D5 | The organization emphasizes permanence and stability. Efficiency, control, and smooth operations are important. |
| Criteria of success | A6 | The organization defines success on the basis of the development of human resources, teamwork, employee commitment, and concern for people. |
| | B6 | The organization defines success on the basis of having the most unique or newest products. It is a product leader and innovator. |
| | C6 | The organization defines success on the basis of winning in the marketplace and outpacing the competition. Competitive market leadership is key. |
| | D6 | The organization defines success on the basis of efficiency. Dependable delivery, smooth scheduling, and low-cost production are critical. |

Four alternative answers are offered to each question (A—clan, B—adhocracy, C—market and D—hierarchy), each answer corresponding to one of the four types of organizational culture. Four alternative answers within each question carry a total of 100 points, and they are distributed in the way the respondent gives the most points to the answer that best corresponds to the culture of the organization in which they work, gives fewer points to the next closest answer and so on, taking into account that the total number of points for each dimension must be 100. The completed questionnaires then go through statistical processing, and the final results should give an assessment of the existing organizational culture and show what it should be according to the wishes of employees in the organization. CVF is one of the most influential and widely used models in the field of organizational culture research compared to other models and scales. CVF and organizational culture assessment instrument (OCAI) have better validation and reliability and are very convenient for practical operations [21].

In Table 4, the formulation of questions for each dimension are presented.

First, descriptive statistics for each dimension are presented (see Table 5).

From Table 5, it can be seen that averages for questions in different dimensions of the organizational culture, vary from 19.8 to 31.3. When looking for each dimension separately, it can be seen that for dominant characteristics, respondents see their organization as a very personal place (M = 28, SD = 16.62) or as a very controlled and structured place (M = 27.55, SD = 15.02), rather than as a very result-oriented place (M = 24.33, SD = 12.46) or a very dynamic and entrepreneurial place (M = 20.21, SD = 11.93). Regarding the organizational leadership, respondents see their organization more as generally considered to exemplify mentoring, facilitating, or nurturing (M = 28.32, SD = 17.74) or as generally considered to exemplify coordinating, organizing, or smooth-running efficiency (M = 27.29,

SD = 15.01), rather than as generally considered to exemplify a no-nonsense, aggressive, results-oriented focus (M = 23.24, SD = 11.58) or as generally considered to exemplify entrepreneurship, innovation, or risk taking (M = 21.25, SD = 13.54). When looking at the management of employees, respondents tend to see their organization as characterized by teamwork, consensus, and participation (M = 28.94, SD = 17.67) or as characterized by security of employment, conformity, predictability, and stability in relationships (M = 27.93, SD = 15.31), rather than characterized by hard-driving competitiveness, high demands, and achievement (M = 23.44, SD = 12.35) or characterized by individual risk taking, in-novation, freedom, and uniqueness (M = 19.81, SD = 12.22). As for organizational glue, respondents see their organization more as running on loyalty and mutual trust (M = 29.42, SD = 18.01), rather than as formal rules and policies (M = 26.58, SD = 15.26), as having emphasis on achievement and goal accomplishment (M = 24.25, SD = 13.72), or as com-mitment to innovation and development (M = 19.98, SD = 12.72). When looking into strategic emphasis, respondents see their organization as the one that emphasizes human development (M = 30.58, SD = 17.29), rather than as the one that emphasizes acquiring new resources and creating new challenges (M = 23.62, SD = 13.94), as the one that emphasizes permanence and stability (M = 23.52, SD = 13.86), or as the one that emphasizes competitive actions and achievement (M = 22.29, SD = 13.01). Regarding criteria of success, respondents see their organization as the one that defines success on the basis of the development of human resources, teamwork, employee commitment, and concern for people (M = 31.32, SD = 17.34), rather than as the one that defines success on the basis of efficiency (M = 24.49, SD = 15.85), the one that defines success on the basis of having the most unique or newest products (M = 23.58, SD = 14.25), or as the one that defines success on the basis of winning in the marketplace and outpacing the competition (M = 20.62, SD = 13.02).

The OCAI questionnaire should not be expected to determine the ideal but only the dominant type of organizational culture, whether it is about current or desired culture. An organization rarely has only one type of culture; much more often, there is a combination of all four types of organizational culture. There is no ultimate "best" organizational culture, although a certain type may be better than others in certain situations.

First, for each of the six dimensions of organizational culture, the dominant type of organizational culture was determined. Results can be found in Table 6.

From Table 6, it can be seen that for all six dimensions of organizational culture, the dominant type of the organization culture is the clan culture: from 35% (for organizational leadership) to 47% (for criteria of success) of respondents chose the clan culture as the dominant organizational culture for all six dimensions of organizational culture. The second most dominant organizational culture, selected by respondents, was the hierarchy culture: between 18.6% (for strategic emphasis; in this case, more respondents, 20.1%, selected the adhocracy culture as the dominant type of organizational culture) and 33.6% (for the dominant characteristics and management of employees), the hierarchy culture was selected as the dominant type of organizational culture. Market culture was selected as the third most dominant organizational culture: between 12.1% (for criteria of success; in this case, adhocracy culture was selected by 17.8% respondents as the dominant culture) to 18.7% (for dominant characteristics) of respondents, the market culture was selected as the dominant type of organizational culture. Adhocracy culture was almost always selected in fewer cases as the dominant organizational culture: between 11.2% (for management of employees) and 20.1% (for strategic emphasis) of respondents, the adhocracy culture was selected as the dominant type of organizational culture.

In the next step, for each respondent, the dominant organizational culture was selected, according to the prevalence of responses for each characteristic of the organizational culture. Results are presented in Table 7.

From Table 7, it can be seen that majority of respondents (43.1%) work in an organi-zation with clan culture, 32.2% of them work in an organization with hierarchy culture, 12.4% of them work in an organization with market culture and 12.3% of them work in an organization with adhocracy culture.

**Table 5.** Descriptive statistics for the dimensions of organizational culture.

| Dimension of Organisational Culture | | Mean | Std. Deviation | Skewness | Kurtosis | Minimum | Maximum |
|---|---|---|---|---|---|---|---|
| Dominant characteristics | A1 | 27.9954 | 16.61635 | 0.762 | 0.897 | 0.00 | 100.00 |
| | B1 | 20.2071 | 11.93391 | 0.900 | 1.326 | 0.00 | 70.00 |
| | C1 | 24.3309 | 12.46101 | 1.047 | 3.210 | 0.00 | 100.00 |
| | D1 | 27.5496 | 15.01851 | 0.753 | 0.776 | 0.00 | 100.00 |
| Organisational leadership | A2 | 28.3233 | 17.74268 | 0.980 | 1.128 | 0.00 | 100.00 |
| | B2 | 21.2490 | 13.54387 | 1.238 | 2.773 | 0.00 | 100.00 |
| | C2 | 23.2390 | 11.57717 | 0.680 | 1.736 | 0.00 | 99.00 |
| | D2 | 27.2921 | 15.01030 | 0.842 | 1.048 | 0.00 | 94.00 |
| Management of employees | A3 | 28.9448 | 17.66831 | 0.772 | 0.753 | 0.00 | 100.00 |
| | B3 | 19.8095 | 12.22323 | 1.134 | 2.640 | 0.00 | 96.00 |
| | C3 | 23.4425 | 12.35102 | 1.025 | 2.857 | 0.00 | 99.00 |
| | D3 | 27.9297 | 15.31041 | 0.696 | 0.776 | 0.00 | 100.00 |
| Organisation glue | A4 | 29.4150 | 18.01216 | 0.883 | 0.730 | 0.00 | 100.00 |
| | B4 | 19.9776 | 12.72386 | 1.063 | 2.540 | 0.00 | 96.00 |
| | C4 | 24.2508 | 13.72305 | 1.269 | 3.150 | 0.00 | 99.00 |
| | D4 | 26.5821 | 15.26070 | 0.784 | 0.840 | 0.00 | 100.00 |
| Strategic emphasis | A5 | 30.5797 | 17.28702 | 1.139 | 1.948 | 0.00 | 100.00 |
| | B5 | 23.6243 | 13.93728 | 1.115 | 1.788 | 0.00 | 70.00 |
| | C5 | 22.2892 | 13.00967 | 1.017 | 2.547 | 0.00 | 100.00 |
| | D5 | 23.5167 | 13.86069 | 1.090 | 2.601 | 0.00 | 100.00 |
| Criteria of success | A6 | 31.3161 | 17.34371 | 0.896 | 1.049 | 0.00 | 100.00 |
| | B6 | 23.5751 | 14.25022 | 1.374 | 3.871 | 0.00 | 100.00 |
| | C6 | 20.6151 | 13.02443 | 1.489 | 4.223 | 0.00 | 100.00 |
| | D6 | 24.4938 | 15.84582 | 1.429 | 3.274 | 0.00 | 100.00 |

*4.2. Testing Research Hypotheses*

In the following, research hypotheses were tested, using CHAID analysis. The dominant organizational culture was inserted as dependent variables, while respondents' characteristics (gender, age, highest level of education, years of working experience and type of employment) and companies' characteristics (type of company and region of the company) were inserted as independent variables. Results can be seen in Figure 2.

The algorithm correctly classified 54.8% of all cases. Six groups of respondents emerged from the analysis.

The largest group comprises 410 (32.9%) of respondents. They work in a hotel in the south of the country. The majority of them, 57.6%, work in an organization with hierarchy culture, 26.6% of them work in an organization with clan culture, 10.5% of them work in an organization with market culture and 5.4% of them work in an organization with adhocracy culture.

The second group is formed of 241 (19.3%) of respondents. They work in a hotel in the north or center of the country. 39.8% of them work in an organization with hierarchy culture, 32% of them work in an organization with clan culture, 16.6% of them work in an organization with adhocracy culture, and 11.6% of them work in an organization with market culture.

**Table 6.** Dominant type of organizational culture.

| Dominant Type of Organisational Culture | Clan Culture | | Hierarchy Culture | | Adhocracy Culture | | Market Culture | |
|---|---|---|---|---|---|---|---|---|
| **Dimension** | **f** | **f%** | **f** | **f%** | **f** | **f%** | **f** | **f%** |
| Dominant characteristics | 475 | 36.0 | 154 | 11.7 | 246 | 18.7 | 443 | 33.6 |
| Organisational leadership | 468 | 35.0 | 207 | 15.5 | 215 | 16.1 | 449 | 33.5 |
| Management of employees | 506 | 37.6 | 150 | 11.2 | 237 | 17.6 | 451 | 33.6 |
| Organisation glue | 482 | 36.8 | 173 | 13.2 | 243 | 18.6 | 411 | 31.4 |
| Strategic emphasis | 475 | 45.1 | 212 | 20.1 | 170 | 16.1 | 196 | 18.6 |
| Criteria of success | 472 | 47.0 | 179 | 17.8 | 122 | 12.1 | 232 | 23.1 |

**Table 7.** Organizational culture in respondents' companies.

| Dominant Type of Organisational Culture | f | f% |
|---|---|---|
| Clan culture | 537 | 43.1 |
| Hierarchy culture | 153 | 12.3 |
| Adhocracy culture | 155 | 12.4 |
| Market culture | 401 | 32.2 |

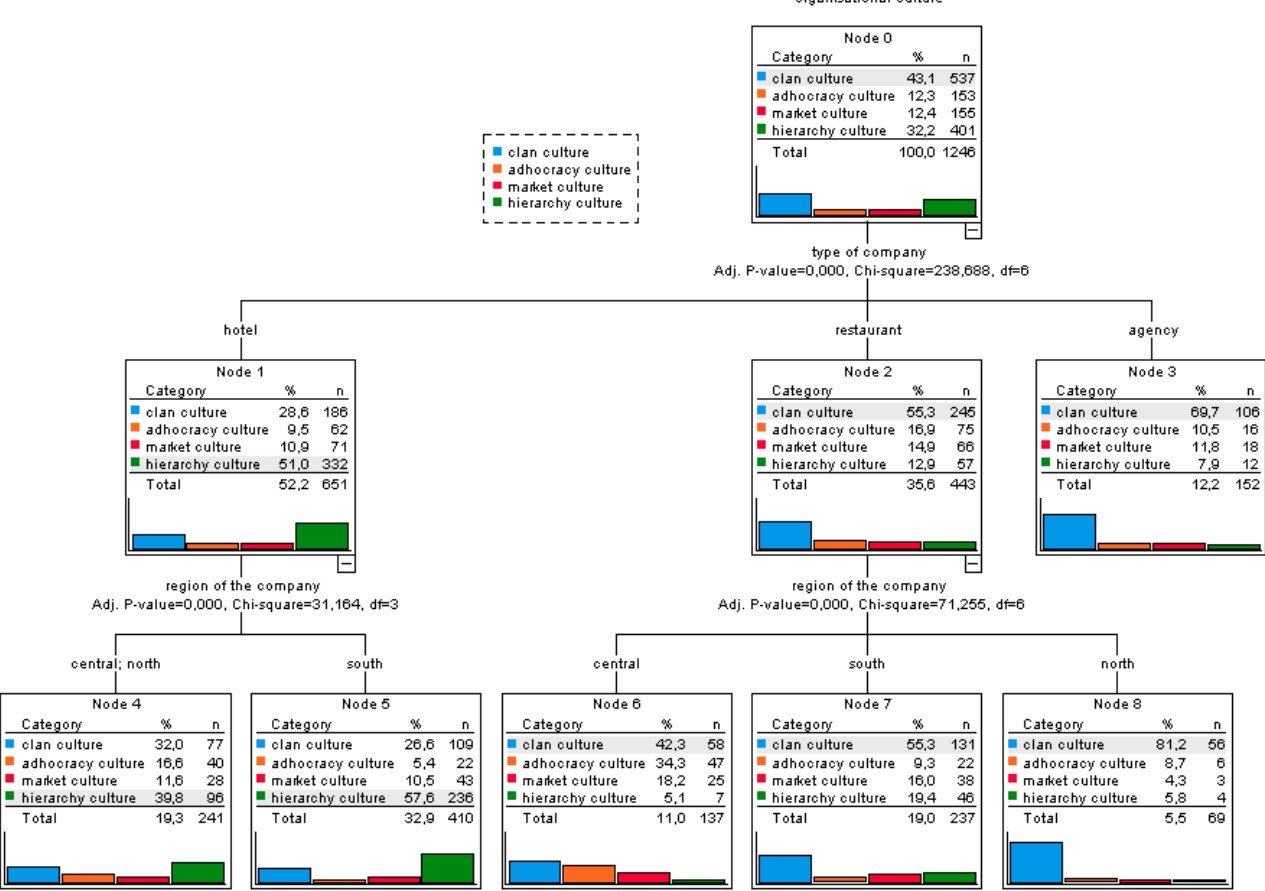

**Figure 2.** The influence of respondents' and companies' characteristics on the dominant type of organizational culture.

The third group is formed of 237 (19%) respondents. They work in a restaurant in the south of the country. For the majority of them (55.3%), the dominant culture in their organization is the clan culture, for 19.4% of them it is the hierarchy culture, for 16% of them it is the market culture and for 9.3% of them it is the adhocracy culture.

In the fourth group, there are 152 (12.2%) of the respondents. They work in a travel agency. 69.7% of them work in an organization with clan culture, 11.8% of them work in an organization with market culture, 10.5% of them work in an organization with adhocracy culture and 7.9% of them work in an organization with hierarchy culture.

The fifth group is formed of 137 (11%) of the respondents. They work in a restaurant in the central part of the country. 42.3% of them work in an organization with clan culture, 34.3% of them work in an organization with adhocracy culture, 18.2% of them work in an organization with market culture, and 5.1% of them work in an organization with hierarchy culture.

The smallest group is formed of 69 (5.5%) of the respondents. They work in a restaurant in the north of the country. The majority, 81.2%, of them work in an organization with clan culture, followed by 8.7% of them who work in an organization with adhocracy culture, 5.8% of them who work in an organization with hierarchy culture and by 4.3% of them who work in in organization with market culture.

## 5. Discussion

The main finding of our research is the fact that clan culture, as the dominant form of organizational culture, is most often found in restaurants in the north of Montenegro and then in travel agencies and restaurants in the south of Montenegro. The culture of hierarchy can most often be found in hotels in the south of Montenegro, which confirms previous research [91]. Hotels in northern and central Montenegro have a market or clan culture as the dominant type of organizational culture, while restaurants in central Montenegro have a clan or adhocratic organizational culture.

In addition, considering the gender, age, level of education, years of work experience and type of employment of the respondents, their characteristics and understandings differ and they identify with the prevailing culture in a different way. This represents an important result of our study as it confirms hypothesis 1; that is, the characteristics of the respondents (gender, age, highest level of education, years of work experience and type of employment) affect the perception of organizational culture. This is supported by the findings of Schein [2] who emphasized that organizational culture exerts its influence through shaping the behavior of organizational members, which depends on their perception of organizational culture. Furthermore, the experience is used as a basis for learning and understanding the organizational culture in one's own organization and the way in which employees feel and perceive the culture in their organization. This is in accordance with similar researches [4], which indicate that knowledge of the organizational culture shapes the causes and consequences of people's behavior in the organization. The need for belonging, which each person carries in themselves to a greater or lesser extent, can only be satisfied by the members of organization if they identify with the organization. Higher level of education, age, years of work experience and type of employment influence a better understanding of the organizational culture and its cognition, which makes it possible for employees to form a way of thinking, reacting and behaving in accordance with the assumptions and values on which the organizational culture rests, which further means that they are identified with the existing practice of the organization and thus contributed to its success, i.e., aligned with the culture of the organization, which is in accordance with the research of Justwan et al. [40], whose findings show that organizational culture affects the relationship between hotel owners and employees by building trust between them and influencing organizational commitment and the performance of the employees. Culture determines the quality of working environment and professional performance [17].

In hypothesis 2, we tested the relationship between a company's characteristics (type of the company and the region of the company) and perception of organizational culture.

Our findings point to the apparent connection of organizational culture with the type and size of tourist companies in the Montenegrin tourism and hospitality sector. This connection is reflected in the following: small and medium-sized hotels, restaurants and tourist agencies are still dominated by clan culture, which can be explained by the fact that these are small collectives with strong interpersonal ties and mutual respect; the hierarchical structure of management here is very simple, and sometimes there are family ties between the employees and the owner of the hotel, restaurant or agency. This finding is consistent with the study of Denison and Spreitzer [81], which indicates that clan culture is typical for small organizations, which behave more like families and especially value commitment to the organization and loyalty. On the other hand, in large organizations, in this case large hotels, the hierarchy culture is the most logical outcome of the organizational culture, since this type of culture is best harmonized with their distinctly hierarchically organized organizational structure. This finding is in accordance with the study of Zeng and Luo [1] which indicates that hierarchy culture is typical for large organizations, since this culture is characterized by an organizational culture model based on clearly defined hierarchical levels, bureaucracy and process formalization.

In relation to the regional distribution of Montenegrin hotel companies, the findings indicate that the culture of hierarchy is present only in hotels located in the southern tourist region. We interpret this result as a consequence of the uneven economic development of Montenegro, whose main characteristic is the existence of a developed south and much less developed north. This disproportion is especially present in the tourism sector, which is the main branch of the Montenegrin economy and the backbone of its overall economic development. This is in accordance with the findings of Janićijević, and Canning et al. [28,33], whose findings imply that specificities of a country are important and that they form the difference between a geographic and a tourist region, because a tourist region is characterized by the dominant development of tourist activity. Therefore, the highly profit-oriented hierarchy culture can be considered a logical development process, which can be linked to a higher degree of formalization of the process.

According to the level of tourism development, the largest number of hotels is concentrated in the southern tourist region of Montenegro, many of which are mostly owned by foreign investors [92]. The multinational hotel corporations that are now present in the tourism market of Montenegro, and which operate on the principles of franchise and the brand they represent [93], have brought standards that are very strict and structured and a clear hierarchy of process formalization, aimed at achieving greater business efficiency, which is in accordance with research of Yan et al. [94], which indicates that a higher degree of tourism development in a certain region attracts multinational corporations that aspire to greater efficiency and a high hierarchy of business. For this reason, the culture of hierarchy is the most logical outcome in the southern region of Montenegro and in hotel companies, since this type of culture is best aligned with their highly hierarchically organized organizational structure and high-profit business orientation which confirms previous research [35]. We came to the conclusion that the central, and especially the northern, region lag far behind the southern tourist region and these enterprises are still dominated by clan culture, which can be explained by the fact that these are small collectives with strong interpersonal ties and mutual support. Furthermore, research has shown that the clan culture has remained dominant in restaurant companies and travel agencies. The explanation for this result should be found in the fact that restaurants and travel agencies in Montenegro are mostly owned by local entrepreneurs and that, as a rule, these are smaller collectives in which there is a family atmosphere in which all full-time employees know each other and help each other.

Based on the results of the present research, we can state that the remaining two types of organizational culture, i.e., market culture and adhocracy culture, are present in negligible numbers in tourist companies in Montenegro. Namely, they are not nearly as present in domestic tourist companies as clan culture and hierarchy culture. These

results are aligned with the propositions of dominant characteristics of different types of organizational cultures [95].

## 6. Conclusions

Organizational culture has long been the subject of numerous scientific papers in countries with developed economies [10], and this research significantly contributes not only to its better understanding but also in terms of tangible practical results. The connection between theory and practice is fully expressed here since the building of a strong and stable organizational culture is a key prerequisite for gaining and maintaining the competitive advantage of economic organizations, but is equally important for the successful operation of all organizations, regardless of their field of activity [96]. We did not find in the literature any research that analyzes the characteristics of organizational culture in tourism and hospitality companies in underdeveloped countries and countries in transition. Accordingly, due to inadequate knowledge of organizational culture in transition economies, this study contributes to knowledge obtained from previous studies and, therefore, deserves attention because we expanded current findings as previous studies did not consider the regional component. In this study, we conducted research on the characteristics of organizational culture in tourism enterprises in Montenegro, which is a developing country and has gone through the transition process in the economic sector.

When forming the sample of research, which included 134 tourist companies, we took into account that the selected companies best represent the structure of the Montenegrin tourism industry. Hotels, restaurants and tourist agencies are consistently represented, in proportion to their number, size, professional and age structure of the employees and regional affiliation. In doing so, we took into account the regional specifics of Montenegro, which are manifested in the existence of three tourist regions, of which the south is extremely developed, the north very undeveloped, and the central is somewhere in the middle [97]. Unlike the study of Santiago et al. [7] that conducted research of organizational culture only in high-category hotels, which is why the results could not be generalized to the hospitality industry, in this study we also took into account hotels of lower categories, thus completing the existing knowledge and enabling the generalization of the results in the hospitality sector. Moreover, in addition to hotels and restaurants, we included tourist agencies as promoters and agents of tourist travels and conceived a comprehensive approach to researching organizational culture in the tourism industry. Furthermore, starting from understanding that perception and experience are basically considered subjective phenomena and are defined by an individual's characteristics [13]. In this study, we analyzed perception of organizational culture on an individual level, which complements the current literature that mainly analyzes organizational culture on a company level, ignoring individuals' perception.

### 6.1. Theoretical Contribution

Regarding the theoretical contribution, this paper shows and systematizes theoretical knowledge about organizational culture and its dominant types, as well as correlation of organizational culture, individual characteristics and a company's characteristics. In that context, this study is among the first that examines the relationship between individual characteristics of the respondent and performance of the tourism company on one side, and organizational culture on the other.

This study points out the importance of a respondent's individual characteristics in the perception of organizational culture. More precisely, individual characteristics as subjective phenomena are based on values, beliefs and behavior norms, which are from national culture being integrated into organizational culture. Taking into account previous studies that examined the effects of an individual's characteristics, including age, level of education and work experience, on their dedication to organization [18], in this study we expanded current findings, taking into account organizational culture as a variable that significantly influences employees' behavior, primarily in the sense of

interpersonal relationships, socialization and their dedication to organization. Secondly, all literature so far provides a comprehensive review on effects of national culture on organizational culture, and therefore this study gives a special contribution by showing significant influences between regional culture, as specific segment of overall culture on the national level, and organizational culture on the level of tourism company. This study is also among the first that deals with the regional influence on organizational culture in transitional economies, with special regard on former Yugoslavia countries. Namely, Montenegro is a country that has been undergoing a process of transformation of the economic system, social order and value system since the 1990s [26]. The biggest change in this process occurred in the ownership and organizational structure of tourism companies, which have been transformed from self-governing socially owned labor organizations into various forms of private companies with the participation of domestic and foreign capital [27]. The largest number of hotels is concentrated in the southern tourist region of Montenegro, many of which are mostly owned by foreign investors [98]. Compared to previous studies that mainly focused to examine organizational culture in developed countries, the results of this study complement and expand current knowledge, delivering findings from underdeveloped countries, i.e., developing countries, with specificity of geographical area, in other words, its regional distribution. Thirdly, this study points out the importance of tourism company characteristics as the most significant determinant of organizational culture. In other words, depending on the type of tourism company, i.e., size and level of its development, different types of organizational culture will be generated. Moreover, unlike previous studies that examined only hotel enterprises, within this study we developed measurements of organizational culture in the context of tourism sector with focus on hotel enterprises, restaurants and travel agencies, and by doing this we filled in the gap in the literature, given that previous literature [9] discovered that there is an important correlation between corporate cultures and companies' performances.

In this study, we used the competing values framework (CVF) developed by Cameron and Quinn as one of the most influential and widely used models in the field of research of organizational culture [21]. The model recommends integration of corporate culture and company's performance. In that regard, this study examined the correlation between tourism company characteristics, on one side, and different types of organizational culture, on the other. We have linked the features of tourism enterprises (type, size, regional distribution) with the competing values framework (CVF) on a micro level, providing more details on how characteristics, type of the company and regional distribution influence affect the dominant type of organizational culture, and on a macro level regarding tourist destination, giving a general image of type of organizational culture in Montenegrin tourism economy, which is characterized by strong economic transition. By doing so and by respecting all specificities of tourism economy, we generated knowledge that includes both micro and macro level, and so we contribute to existing literature and help company's better performances.

### 6.2. Practical Implications

From a practical point of view, identifying the correlation between the practice and the perception of organizational culture can help tourism companies to establish their culture and identify potential levers for organizational change, which would enable better business performance. This supports the findings of Tavitiyaman et al. [5], who emphasized the importance of organizational culture in the context of corporate strategy, recognizing that organizational culture has become an important aspect for senior management. In this regard, the findings of this research will serve as a foundation and help managers create a tourism company management or tourist destination management strategy on the micro and macro level. Compared to other studies so far, this study gives insight into specificities of different types of tourism companies and perception of organizational culture, as well as differences that can occur in the dominant type of organizational culture regarding

regional distribution, which will additionally contribute to overall knowledge and help with management of tourism companies.

Another convenient contribution that might be important to human resources managers is to understand and accept employees' individual characteristics, starting from an individual's subjective perception, especially in multinational companies typical for tourism sector, in which employees from different cultures introduce their own values and behavior norms into organizational culture. This is especially important because of their dedication to organization and that will further lead to the effective performance of an organization.

Considering the strategic orientation of the Montenegrin economy to the tourism sector and the significant influence of foreign investors, knowledge of organizational culture in tourism companies in Montenegro is a significant contribution to understanding the specifics of the market and the values and norms on which these companies rest. Therefore, the governments are competing more and more to attract multinational hotel corporations by offering significant stimulants related to domain of tax policy and communal fees, as part of overall efforts. This rapidly becomes the critical instrument in the development of the country whose economical growth is dominantly based on hospitality industry. This raises another variable that has not been examined in this study but would be of great importance for further research, and that is the influence of politics and institutional framework. However, considering the increasing number of privatized enterprises in the tourism sector, attitudes towards the influence of direct foreign investments remain discordant and therefore also the justification of microeconomic policies of the host country.

*6.3. Limitations and Recommendations for Future Development*

This study has certain limitations that can lead to future studies. The results of this study can only be generalized to a selected tourism sector. It would be desirable to include other destinations as well in the future studies because development as contextual factor on the country's level can provide different results. Developed countries are characterized by high innovation [41], which suits adhocracy culture [7], which is in this study present at a very low proportion. In this regard, it would be interesting to conduct the same research in other countries.

The next limitation is that in sense that we did not examine how the study's variables could be led by politics. Accordingly, the influence of politics and institutional framework should also be examined in the future studies, especially because politics-led variables can provide different results, namely the influence of politics in the case of tourism companies and their cooperation with foreign colleagues, where the awareness of socially responsible cooperation is high, it is expected that they will be institutionalized by the culture of socially responsible business and ecological sustainability [51].

The next limitation of this studies reflects in the fact that we did not examine sustainability, i.e., organizational culture that is committed to environment, so called "green culture". According the modern tendencies and characteristics of modern societies [54], which are more and more directed towards ecological awareness, this is a very important variable that should be the focus of the future studies. This study has not taken into account practices that hotels use in order to become more sustainable. Therefore, future studies should examine the influence of organizational culture on sustainability practices and the perception of sustainable performance, and not only on characteristics of the company, as was the case in this study.

In this study, we examined the influence of individual characteristics and company characteristics, including regional distribution, on the perception of organizational culture. The influence of national culture on organizational culture should be examined, especially because the hotel sector is characterized my multinational hotel corporations whose employees implement assumptions, values and beliefs from their own national cultures. Due to the different norms and values of the national culture, the proposal for future studies is to

examine how the differences in national culture are reflected in individuals' characteristics, their values and behavior norms, which they implement in organizational culture.

Furthermore, in this study we examined the perception of organizational culture with regard to employees' individual characteristics, but we did not examine employees' satisfaction depending on dominant types of organizational culture. It would be very beneficial to examine the correlation between different types of organizational culture, employees' satisfaction and productivity in the future studies.

**Supplementary Materials:** The following supporting information can be downloaded at: https://www.mdpi.com/article/10.3390/su15032715/s1, File S1: Questionnaire.

**Author Contributions:** Conceptualization, O.S.; methodology, O.S. and E.P.; software, E.P.; validation, E.P.; formal analysis, E.P.; investigation, O.S. and E.P.; resources, O.S. and E.P.; data curation, E.P.; writing—original draft preparation, E.P., M.L.; writing—review and editing, E.P., O.S., M.L.; visualization, E.P., O.S., M.L. and Đ.P.; supervision, E.P.; project administration, O.S.; funding acquisition, O.S. All authors have read and agreed to the published version of the manuscript.

**Funding:** This research received no external funding.

**Institutional Review Board Statement:** Not applicable.

**Informed Consent Statement:** Not applicable.

**Data Availability Statement:** Not applicable.

**Conflicts of Interest:** The authors declare no conflict of interest. The funders had no role in the design of the study; in the collection, analyses, or interpretation of data; in the writing of the manuscript; or in the decision to publish the results. The authors started this paper when Olivera Simović was a PhD student at the Faculty of Economics, University of Belgrade as a part of her thesis.

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
