# Peer review of "Measuring Organizational Culture in Hotels, Restaurants and Travel Agencies in Montenegro"

_sustainability, doi:10.3390/su15032715_

Round 1

Reviewer 1 Report

Dear Authors!

The manuscript provides a contribution to the research area regarding on the study of the measuring organizational culture in hotels, restaurants and travel agencies.

The title of the manuscript is appropriate. The title reflects the content of the manuscript.

There is a balanced relationship between the objective, content of the manuscript.

However:

- “Leading names in the field of psychology and sociology of the organization:” It would not be superfluous to mention the names of scientists as well, and not only to refer to works.

- It is not clear from the text of the article why a rather high confidence interval of 8% was chosen for the sample calculation.

- It is advisable to separate the discussion and conclusions section.

- The conclusions should be systematized. We consider it superfluous to emphasize the goal in the conclusions.

The manuscript is precise and well-written.

I am grateful to the authors for developing an interesting material.

Best regards,

Author Response

Dear reviewer, 

thank you very much for revieweing our manuscript. Your comments were highly appreciated and it made gain more clarity to our work. We have improved our manuscript as suggested by you:

  • “Leading names in the field of psychology and sociology of the organization:” It would not be superfluous to mention the names of scientists as well, and not only to refer to works. REVISION: Thank you very much for your suggestion. We have added the names of the authors.
  • It is not clear from the text of the article why a rather high confidence interval of 8% was chosen for the sample calculation. REVISION: Thank you very much for your comment. We have added a statement "We increased the margin of error from the standard 5% to 8% so that the size of the representative sample would be within 10% of the total number of tourist companies in Montenegro."
  • It is advisable to separate the discussion and conclusions section. REVISION: Thank you very much for your suggestion. We have separated the discussion and conclusions sections.
  • The conclusions should be systematized. We consider it superfluous to emphasize the goal in the conclusions. REVISION: Thank you very much for your suggestion. We have rewritten the conclusions section.

Reviewer 2 Report

This paper has a detailed discussion on the relationship between organizational culture and company type, which has certain practical significance for enterprise management and development. However, it looks like a technical report and it does not satisfy the publication standards due to many problems, for example, low quality research, bad layout, poor English. Of course, I would like to some of my major concerns.

1. The focus of the article is on the relationship between the type of organizational culture, the type of company and their geographical location. And, the aim of the research is to determine the character of the dominant models of organizational culture in the tourism and hospitality sector of Montenegro. What is the relationship and logic between hypothesis 1 and the research aim of this paperIn other words, does the analysis of Respondents' characteristics have any special meaning or interpretation in this case and its results.

2. The concept of Leadership is introduced in detail in the literature review, but there is almost no relevant explanation in other parts of the article. Is the concept of leadership needed? What, if any, role does this concept play in the text? Does it have any special contributions to the innovation and significance of the article?

3.It is suggested to fill in the name of the author quoted in the article. For example, in line 172, ‘Finding by Alvesson points out…’, rather than ‘Finding by [28] point out…’.

4.There are many errors in the format of the article. For example, the punctuation in line 209, 282 of the text. Please check carefully and revise it again. In addition, Table 1 and Table 4 is not very pretty, please modify it as appropriate.

5.The quantity and quality of references need to be revised again, and the appropriate amount of literature should be deleted.

Author Response

Dear reviewer,

thank you very much for all your valuable comments. We have made necessary revisions, according to your suggestions:

  1. The focus of the article is on the relationship between the type of organizational culture, the type of company and their geographical location. And, the aim of the research is to determine the character of the dominant models of organizational culture in the tourism and hospitality sector of Montenegro. What is the relationship and logic between hypothesis 1 and the research aim of this paper?In other words, does the analysis of Respondents' characteristics have any special meaning or interpretation in this case and its results. REVISION: Thank you very much for your comment. An explanation of hypothesis 1 is given in more detail in the new version of the manuscript.
  1. The concept of Leadership is introduced in detail in the literature review, but there is almost no relevant explanation in other parts of the article. Is the concept of leadership needed? What, if any, role does this concept play in the text? Does it have any special contributions to the innovation and significance of the article? REVISION: Thank you very much for your comment. We have made appropriate modifications in the new version of the manuscript and deleted the concept of leadership from the manuscript.

3.It is suggested to fill in the name of the author quoted in the article. For example, in line 172, ‘Finding by Alvesson points out…’, rather than ‘Finding by [28] point out…’. REVISION: Thank you very much for your valuable comment. We have made the necessary corrections in the new version of the manuscript, according to your suggestion.

4.There are many errors in the format of the article. For example, the punctuation in line 209, 282 of the text. Please check carefully and revise it again. In addition, Table 1 and Table 4 is not very pretty, please modify it as appropriate. REVISION: Thank you very much for your comment. We have deleted Table 1 and modified Table 4. We have also corrected the punctuations.

5.The quantity and quality of references need to be revised again, and the appropriate amount of literature should be deleted. REVISION: Thank you very much for your suggestion. As it can be noted from the new version of the manuscript, references were revised and not necessary references were deleted.

Reviewer 3 Report

Please, see attached document.

Author Response

Dear reviewer,

thank you very much for your valuable comments. In the newest version of the manuscript we have addresses your suggestions as following:

1. In the abstract, the authors need to be focused and state exactly the purpose of the study, brief description of the method adopted for the study, findings, and at least one implications of the study. REVISION: Thank you very much for your suggestion. The abstract was revised according to your comment.
2. The introduction needs to be strengthened to justify why the variables are important to the current organizational environment. Thus, the introduction needs to be more structured. It could be better, then the introduction follows the theme of motivation, a brief story, literature gap, and contribution. It is not a place for the authors to present their findings rather to throw more light on the contribution. REVISION: Thank you very much for your comment. The introduction was revised accordingly.
3. In the literature section, the authors need to follow the journal citation of in-text. Besides, most of the lines are without justification. Also, the hypotheses development is poorly done. The authors should come again to develop well-structured hypotheses to take the readers along, not for the readers to deduce to get understanding. It is about how the characteristics of the company, regional distribution of tourism business, and respondents affect the perception of organizational culture. REVISION: Thank you very much for your comment. The literature review was revised.

4. The study adopted questionnaire survey approach, but its fail to highlight whether the variables were developed or adopted. The authors should let us know all these things. For instance, hypothesis 1 states that respondents’ characteristics influence the perception of organizational culture. Similarly, the study fails to demonstrate how the variables were measured. Hence, I can consider the questionnaire as inappropriate. Besides, the study uses a questionnaire survey and a potential threat of common method bias cannot be
ignored. In addition, why the use of CHAID analysis? Finally, there is no questionnaire attached to the manuscript to assess its suitability for the study. REVISION: Thank you very much for your comment. The questionnaire development was added into the new version of the manuscript. The full questionnaire can be found in the appendix. The use of CHAID analysis was explained in the new version of the manuscript.
5. From the study, the theoretical and practical contributions need to be strengthened. Thus, discussions on theoretical contributions need to be improved. For each contribution, authors need to first display what the prior theory says and then argue how their findings extend the theory. REVISION: Thank you very much for your comment. The contributions were added in the newest version of the manuscript.
6. Also, the authors can create a conclusion section to show the implications for the research by further evaluating how the study variables can be driven by policy and how research findings can fill gaps in the literature. REVISION: Thank you very much for your comment. A discussion section was added in the new version of the manuscript.

Reviewer 4 Report

The topic itself is very interesting, in general the paper is of high quality, though it can be further improved. I would suggest to revise the text and in some cases make the content "more scientific". In the current version, the paper sometimes seems to be too "simple and ordinary".

I would also suggest to change plural (we) to passive voice to moderate the too personal voice through the paper.

There are also some grammar and spelling mistakes or formatting errors e.g. table 4 (extra numbers in the table).

Regarding the conclusions, findings, authors focused much on how their results and findings are in harmony with other authors but they did not highlight how much and why their findings are new and how much they contribute significantly to the topic. I would suggest to emphasize more the unique findings of the paper.

Author Response

Dear reviewer,

thank you very much for all your valuable comments. According to your suggestion, the necessary modifications were made in the new version of the manuscript:

The topic itself is very interesting, in general the paper is of high quality, though it can be further improved. I would suggest to revise the text and in some cases make the content "more scientific". In the current version, the paper sometimes seems to be too "simple and ordinary". REVISION: Thank you very much for your review. In the new version of the manuscript, necessary modifications were made according to your suggestions.

I would also suggest to change plural (we) to passive voice to moderate the too personal voice through the paper. REVISION: Thank you very much for your suggestion. In the new version of the manuscript, the necessary modifications were made.

There are also some grammar and spelling mistakes or formatting errors e.g. table 4 (extra numbers in the table). REVISION: Thank you very much for your comment. The text was checked again and grammar and spelling mistakes were corrected. Table 4 was also corrected.

Regarding the conclusions, findings, authors focused much on how their results and findings are in harmony with other authors but they did not highlight how much and why their findings are new and how much they contribute significantly to the topic. I would suggest to emphasize more the unique findings of the paper. REVISION: Thank you very much for your suggestion. The discussions and conlusions sections were rewritten and separated.

Round 2

Reviewer 3 Report

Please, see the attached document

Author Response

Dear reviewer,

thank you very much for all your valuable comments. We appreciate them very much. We believe, your review made a huge improvement to our paper. Please, find below the answers to your comments.

  1. Please, I will encourage the authors to depart from normal theses style of writing, since this is an academic paper and need to be structured well. The problem is with the entire paper. Authors need to be focused and pinpoint the exact issues that may kindle readers to develop interest in the entire story. Please, take your time to develop the story. 

ANSWER: Thank you very much for your comment. We have substantially improved the structure of the paper, in order for it to be more accademically sound. A story was developed through all the paper.

2. Besides, authors need to pick or read current papers published in this journal to serve as a guide. The format is very important. (e.g., 1.0 introduction; 2.0 literature review, 2.1 Hypotheses development; 3.0 methodology, 3.1 sample and data collection, 3.2 variable and measurement; 4.0 Results, 4.1 descriptive statistics and correlation, 4.2 testing of hypotheses 5.0 Discussion and conclusion, 5.1 theoretical implication, 5.2 managerial implication, 5.3 limitation and future studies; 6.0 Conclusion

ANSWER: Thank you very much for your comment. The paper was formatted according to your suggestion.
3. In the abstract, the authors need to be focused and state exactly the purpose of the study, brief description of the method adopted for the study, findings, and at least one implications of the study.

ANSWER: Thank you very much for your comment. We have re-written the abstract according to your suggestion. At this point, thanks to your comment, the abstract is more clear and presents the main points of the paper.
4. The introduction needs to be strengthened to justify why the variables are important to the current organizational environment. Thus, the introduction needs to be more structured. It could be better, then the introduction follows the theme of motivation, a brief story, literature gap, and contribution. It is not a place for the authors to present their findings rather to throw more light on the contribution.

ANSWER: Thank your for your comment. The introduction was significantly modified, putting emphasis on details you provided to us in your review.
5. In the literature section, the authors need to follow the journal citation of in-text. Besides, most of the lines are without justification. Also, the hypotheses development is poorly done. The authors should come again to develop well-structured hypotheses to take the readers along, not for the readers to deduce to get understanding. It is about how the characteristics of the company, regional distribution of tourism business, and respondents affect the perception of organizational culture.

ANSWER: Thank you very much for your comment. We have corrected the citing and referencing. We have also checked the wording and the sentences in the paper. As of your suggestion, research hypotheses were developed, in concordance with previous literature suggestions.
6. From the study, the theoretical and practical contributions need to be strengthened. Thus, discussions on theoretical contributions need to be improved. For each contribution, authors need to first display what the prior theory says and then argue how their findings extend the theory.

ANSWER: Thank you very much for your comment. We have added theoretical and practical contributions of the research, in the conclusions section. As of your valuable suggestion, we have added the explanation of each contribution with previous theories.
7. Also, the authors can create a conclusion section to show the implications for the research by further evaluating how the study variables can be driven by policy and how research findings can fill gaps in the literature.

ANSWER: Thank you for your comment. We have developed a new, conclusions section and future research section, in which we explained the implications of obtained results and how to improve the research in future.
8. Besides, authors need to create a section for the limitations and future direction. The limitations seem a bit generic and under-developed. For example, which other variables could be interesting to test that it was not possible in this one? Which contextual country level factors could matter, which make it interesting to test in other countries? The more details you can provide here, the more useful for other interested researchers.

ANSWER: Thank you very much for your comment. We have added a limitations and future research section, according to your suggestion. In this section, we have answered the questions you proposed.
9. Also, a copy editing is needed. A native English speaker need to go through the entire manuscript to polish some sentences. 

ANSWER: Thank you very much for your comment. A native English speaker checked the manuscript, improving the overall English of the paper.

Round 3

Reviewer 3 Report

Please, address these minor issues.

1. In sustainability the abstract should be 200 words. Besides, it should be unstructured. 

2. Authors should work on all Tables in the text to make them clear.

Author Response

Dear Reviewer,

thank you very much for reading the corrected manuscript we have submitted. We really appreciate the work you have done. In the following, we address the minor issues you suggested.

  1. In sustainability the abstract should be 200 words. Besides, it should be unstructured.

ANSWER: Thank you very much for your comment. The abstract was shortened and unstructured.

2. Authors should work on all Tables in the text to make them clear.

ANSWER: Thank you very much for your comment. All the tables were addressed and made more clear.

Happy Holidays!
